

# Coupling a weather model directly to GNSS orbit determination

Angel Navarro Trastoy[1], Sebastian Strasser[2], Lauri Tuppi[1], Maksym Vasiuta[1], Markku Poutanen[3], Torsten Mayer-Gürr[2], and Heikki Järvinen[1]

[1]Institute for Atmospheric and Earth System Research / Physics, Faculty of Science, University of Helsinki, Finland
[2]Institute of Geodesy, Graz University of Technology, Austria
[3]Finnish Geospatial Research Institute, National Land Survey of Finland

**Correspondence:** Heikki Järvinen (heikki.j.jarvinen@helsinki.fi)

**Abstract.**

Neutral gas atmosphere bends and delays propagation of microwave signals in satellite-based navigation. Weather prediction models can be used to estimate these effects by providing 3-dimensional refraction fields to estimate signal delay in the zenith direction and determine a low-dimensional mapping of this delay to desired azimuth and elevation angles. In this study, a
global numerical weather prediction model (Open Integrated Forecasting System (OpenIFS) licensed for Academic use by the European Centre for Medium-Range Weather Forecast) is used to generate the refraction fields. The ray-traced slant delays are supplied as such – in contrast to mapping – for an orbit solver (GROOPS (Gravity Recovery Object Oriented Programming System) software toolkit of the Technical University of Graz) which applies the raw observation method. Here we show that such a close coupling is possible without need for major additional modifications in the solver codes. The main finding
here is that the adopted approach provides a very good *a priori* model for the atmospheric effects on navigation signals, as measured with the midnight discontinuity of Global Navigation Satellite System (GNSS) satellite orbits. Our interpretation is that removal of the intermediate mapping step allows to take advantage of the local refraction field asymmetries in the GNSS signal processing. Moreover, the direct coupling helps in identifying deficiencies in the slant delay computation because the modelling errors are not convoluted in the precision-reducing mapping. These conclusions appear robust, despite the relatively
small data set of raw code and phase observations covering the core network of 66 ground-based stations of the International GNSS Service over one-month periods in December 2016 and June 2017. More generally, the new configuration enhances our control of geodetic and meteorological aspects of the orbit problem. This is pleasant because we can, for instance, regulate at will the weather model output frequency and increase coverage of spatio-temporal aspects of weather variations. The direct coupling of a weather model in precise GNSS orbit determination presented in this paper provides a unique framework for
benefiting even more widely than previously the apparent synergies in space geodesy and meteorology.

## 1 Introduction

Refraction in the neutral gas atmosphere bend and delay global navigation system satellite (GNSS) signals (Bevis et al., 1992). These atmospheric effects cannot be deduced based on GNSS measurements alone because the signal propagation is identical for all frequencies typically applied in GNSS. Some auxiliary information is therefore needed to correct these





tropospheric effects (Guerova et al., 2016). Global numerical weather prediction (NWP) systems are currently the best source of such support since they ingest tens of millions of observations every day from global meteorological observing systems and constantly update the atmospheric state with the latest data (Bauer et al., 2015).

State-of-the-art in correcting the tropospheric effects in precise orbit determination (POD) of GNSS is based on the use of mapping functions (e.g., Böhm et al., 2006; Niell, 1996; Rocken et al., 2001; Zus et al., 2014). These are designed to encapsulate

essential atmospheric effects by a small number of parameters. The functional form of the Vienna mapping function (VMF3; Landskron and Böhm, 2018), for instance, represents the tropospheric delay effect with three parameters and the azimuthally asymmetric part with two additional parameters. This concise representation makes mapping functions generally easy to apply and exchange but also truncates atmospheric information contained in weather models. In particular, they are not designed to represent azimuthal asymmetries which can be significant, depending on the site and weather of the day (Eresmaa et al.,

2007, 2008a).

This article presents a new configuration where a NWP model is directly coupled with a GNSS orbit solver, thus effectively omitting the use of mapping functions. Our aim is to study the impact of loss-less use of weather model data in orbit determination. To our knowledge, this has not been attempted before on global scale but there are some useful early precedents. Nordman et al. (2007, 2009) applied directly the tropospheric slant-path corrections derived from a limited area weather model

to a regional network solution of the Global Positioning System (GPS) receivers and reported a neutral-to-positive impact. A fairly similar approach was adopted in Hobiger et al. (2008) who concentrated on precise point positioning in a regional GPS network. Finally, Eriksson et al. (2014) applied tropospheric corrections directly and noticed a sizable improvement in a very long baseline interferometry (VLBI) application by omitting the mapping step. Promisingly, Zus et al. (2014) concluded that computational performance of the best codes to compute tropospheric slant-path corrections is no longer a limiting factor in

GNSS processing to side-line the mapping step.

This article is organized as follows. Data and methods are presented in Section 2, results in Section 3 followed by a discussion in Section 4 and the conclusions in section 5.

## 2 Data and methods

This section explains the data and methods applied in the new configuration to solve the GNSS orbit problem. It should be

noted that all input data and different solver codes reside in the same super-computer, that is CSC – the IT Center for Science in Finland (for system specifications, please see CSC, 2021). Different work flow components (running the weather model, production of the slant delay data, and solving the GNSS orbits) are interlaced but managed as separate tasks. Also, all these components are assembled together as such with no lengthy fine tuning undertaken for obtaining the highest possible performance – neither in terms of computational optimization nor solution accuracy. This constitutes in other words a benchmark

system where to build-on further improvements.





## 2.1 Precise orbit determination

The orbit determination process applied in this study is based on the raw observation approach (Schönemann et al., 2011; Schönemann, 2014) and follows the strategy detailed in Strasser et al. (2019). Raw code and phase observations from a network of 66 well-distributed ground-based stations of the International GNSS Service (IGS; Johnston et al., 2017) to the GPS satellite constellation are utilized to determine a set of geodetic parameters in an iterative least-squares adjustment. These observations are processed in daily batches at a 30-second sampling period using GROOPS (Mayer-Gürr et al., 2020), which is an open-source software package for GNSS processing and gravity field determination developed at Graz University of Technology. The observations are connected to the parameters via the code and phase observation equations (e.g., Hauschild, 2017), which encompass corrections for various effects, one of which is the tropospheric influence.

The geometry is contained in the observation equations as the range between satellite and station positions. Satellite orbits are numerically integrated over 24 hours based on force models, such as Earth's gravity field, tidal forces, and radiation pressure (cf. Strasser et al., 2019, for a complete list). These dynamic orbits are then fitted to the observations in the least-squares adjustment by estimating their initial position and velocity as well as a set of solar radiation pressure parameters (Arnold et al., 2015), as this force cannot be modeled adequately in advance. In addition, small instantaneous velocity changes, pseudo-stochastic pulses (e.g., Hugentobler and Montenbruck, 2017), are estimated at the center of each 24-hour orbit arc to consider orbit modeling deficiencies.

Next to the orbit parameters, various other parameters are estimated in the least-squares adjustment. These comprise static station positions, Earth orientation parameters, epoch-wise clock errors and constant signal biases at each receiver and satellite, the ionospheric slant total electron content (STEC) per group of observations between a receiver and satellite at one epoch, and phase ambiguities. The ambiguities are resolved to integer values during processing.

Following Petit and Luzum (2010), the tropospheric slant delay for a line-of-sight observation between a satellite and receiver is

$$T_{SD} = m_h(e)D_{zh} + m_w(e)D_{zw} + m_g(e)[G_N \cos a + G_E \sin a]. \tag{1}$$

Here, $e$ and $a$ are the elevation and azimuth angles at the receiver antenna, $D_{zh}$ is the zenith hydrostatic delay, $D_{zw}$ is the zenith wet delay, and $G_N$ and $G_E$ are the horizontal delay gradients in north-south and east-west directions (the delays are expressed in meters). The mapping functions ($m_h$, $m_w$, and $m_g$) map the delays from zenith to the line-of-sight elevation. The zenith delay mapping function is

$$m_{h,w}(e) = \frac{1 + \dfrac{a}{1 + \dfrac{b}{1+c}}}{\sin e + \dfrac{a}{\sin e + \dfrac{b}{\sin e + c}}}, \tag{2}$$

where $a$, $b$, and $c$ are mapping coefficients (Herring, 1992; Niell, 1996). The gradient mapping function is

$$m_g(e) = \frac{1}{\sin e \, \tan e + C} \tag{3}$$





with $C = 0.0031$ for the hydrostatic effect and $C = 0.0007$ for the wet effect (Chen and Herring, 1997).

In the default system, *a priori* tropospheric corrections are based on VMF3 operational data (Landskron and Böhm, 2018). The model provides zenith hydrostatic, zenith wet, hydrostatic gradient, and wet gradient delays as well as corresponding mapping function coefficients ($a$, $b$, $c$) on a global 1x1° grid at a 6-hour sampling period. Values for a specific station location and point in time are bilinearly interpolated from the surrounding grid points in the space domain and linearly interpolated in time. A height correction from orographic height to station height is applied as well (cf. Strasser et al., 2019).

In the experiment, on the other hand, OpenIFS-based slant delay sky-views (see Section 2.3) are used to determine *a priori* tropospheric delays. These sky-views comprise slant delays at a regular 1x1° azimuth-elevation grid ray-traced directly at the station coordinates with an hourly sampling period. The slant delay for a specific observation is then computed by interpolating the gridded slant delays to the azimuth and elevation of the observation and linearly interpolating in time between the hourly delay values. The interpolation in space domain is done linearly in azimuth direction and using a degree-5 polynomial in elevation direction to properly cover the rapid delay changes at low elevations. Since these slant delays are based directly on a weather model, they include all hydrostatic, wet, and gradient effects and no mapping functions need to be used.

For individual measurements, there is a discrepancy $\Delta T_{SD}$ between the *a priori* tropospheric slant delay and the actual delay affecting the measurement since the models are imperfect. If we assume that $\Delta T_{SD}$ is solely due to troposphere, we can write the tropospheric slant delay discrepancy by using Eq. 1 as

$$\Delta T_r^s = m_h(e)\Delta D_{zh} + m_w(e)\Delta D_{zw} + m_g(e)[\Delta G_N \cos a + \Delta G_E \sin a]. \qquad (4)$$

Assuming that the hydrostatic delay is well modelled ($\Delta D_{zh} = 0$), the residual zenith wet ($\Delta D_{zw}$) and gradient ($\Delta G_N, \Delta G_E$) delay parameters can be set up to consider the discrepancy between modeled and measured tropospheric influence. These station-wise parameters are set up or omitted in this study depending on the analysis. If included, $\Delta D_{zw}$ is parameterized as a degree-1 spline with 2-hourly nodes, while $\Delta G_N$ and $\Delta G_E$ are parameterized linearly over the 24 hours. The mapping functions required for this parametrization are determined using the VMF3 mapping coefficients. The same parametrization is used both in case of VMF3-based or OpenIFS-based *a priori* slant delays since the parametrization is independent of the *a priori* model.

## 2.2 Weather model

OpenIFS is a portable version of the Integrated Forecasting System (IFS) of the European Centre for Medium-Range Weather Forecasts (ECMWF). Essentially, it is a global weather prediction model with identical forecast skill as in the full IFS. Data assimilation is not included in OpenIFS, and thus external initial conditions of atmospheric state (temperature, wind, humidity, and surface pressure) are required. In this study, we use operational atmospheric analyses of ECMWF (Ollinaho et al., 2021) and OpenIFS version 43r3v1 that was part of the operational forecasting system at ECMWF from July 2017 to June 2018 (IFS cycle 43r3; ECMWF, 2019).

We simulate time evolution of the atmospheric state with OpenIFS at horizontal resolution $T_L639$, which corresponds to about 31 km grid spacing at the equator, and at 91 vertical levels. The model top is at 0.01 hPa, which is approximately at the





altitude of 80 km. The model domain thus covers the entire neutral atmosphere of the Earth. The model time step is 15 minutes
and the atmospheric state is output once every hour. For comparison, the ECMWF operational system at the time used 8 km
grid spacing at 137 levels (ECMWF, 2019).

Atmospheric states for the hourly refraction computation for the months of December 2016 and June 2017 are formed as
follows. A sequence of 12-hour forecasts is generated from the 00 and 12 UTC analyses. Thus, an analysis at 00 UTC and
forecasts at 01, 02, ..., 11 UTC, and correspondingly, an analysis at 12 UTC and forecasts at 13, 14, ..., 23 UTC provide
the hourly coverage for one 24-hour period. OpenIFS thus effectively extrapolates the atmospheric analyses to the full hours
between the twice-a-day analysis times.

### 2.3    Slant delays

The tropospheric slant delays are computed with ray tracing based on a least traveltime (LTT) operator (Eresmaa et al.,
2008b). LTT operates on a 2-dimensional plane defined by the satellite and receiver positions and the centre of the Earth.
A 2-dimensional refractivity field is obtained by converting the 3-dimensional OpenIFS fields of atmospheric dry mass and
moisture content to refractivity and interpolating to a desired 2-dimensional plane. Interpolation is bi-linear in horizontal and
assumes exponential refractivity profile in vertical between the model levels.

The LTT algorithm performs ray tracing to construct the path, expressed in polar coordinates, which satisfies the following
system of differential equations:

$$\frac{dr}{ds} = \cos\theta \tag{5a}$$

$$\frac{d\psi}{ds} = \frac{\sin\theta}{r} \tag{5b}$$

$$\frac{d\theta}{ds} = -\sin\theta\left[\frac{1}{r} + \left(\frac{\partial n}{\partial r}\right)_\psi\right] \tag{5c}$$

Here $ds$ is a path element, $r$ is distance from the Earth's centre, $\psi$ is the counterpart for polar coordinates, $\theta$ is the zenith
angle, and $n$ the refractive index. Equations 5 are integrated with the fourth-order Runge-Kutta method starting from the
receiver position ($r_{rec} = R_\oplus + h_{rec}$; $\psi_{rec} = 0$) in the initial direction stated as the geometrical zenith angle of the satellite
($\theta_{rec} = \theta_{geom}$). The integration ends when the satellite altitude is reached ($r_{end} = r_{sat}$). The integration yields a set of points
($r; \psi$) which satisfy these equations.

The total delay on a slanted path results from slowing down of the signal due to refractive index $n > 1$ and increasing signal
path length due to signal bending. The path bending is due to the second term in Eq. 5c. Since geometrical zenith angle is used
as the initial direction, an angular separation appears between the end point of the ray and the satellite location, i.e., $r_{end} = r_{sat}$
but $\psi_{end} \neq \psi_{sat}$. Therefore the calculation of the path and slant delay is repeated by using an updated $\theta'_{rec}$ as follows:

$$\theta'_{rec} = \theta_{geom} - (\psi_{end} - \psi_{sat}) = \theta_{geom} - \Delta\psi \tag{6}$$





In our implementation (Eresmaa et al., 2008b), the final slant delay is a linear combination of these two LTT calculations. This yields an angular difference of the order of $\Delta\psi \approx 10^{-4}$ rad for zenith angle $\theta = 85°$ and $\Delta\psi \approx 10^{-7}$ rad for $\theta = 10°$. Additional accuracy of the starting zenith angle could be obtained with additional iterations.

The effect of Earth flattening was evaluated for the slant delay computation since the LTT algorithm accounts for the Earth oblateness applying the concept of Euler radius. The magnitude of this effect was evaluated by degrading the Earth ellipsoid to a sphere thus changing the 2-dimensional plane projection onto the Earth surface. The effect was found to vary from $-0.5\,\mathrm{cm}$ to $+1.5\,\mathrm{cm}$ depending on the receiver latitude and antenna azimuth angle.

Here the LTT solver is applied such that instead of computing ray paths exactly corresponding individual GNSS measurement geometries, so-called sky-views are generated. They are constructed so that slant delays are calculated for zenith angles from 1 to 89 degrees and azimuth angles from 0 to 359 degrees with one degree increments in both directions, added with one calculation for the zenith delay. The sky-views are computed using the OpenIFS data at every full hour in December 2016 and June 2017 for the 66 selected IGS stations constituting the core network.

## 2.4 Performance metrics

In order to compare the experiments with the default system, the following performance metrics are defined. First, GNSS satellite orbits are determined for 24-hour periods. These 24-hour orbit arcs overlap at midnight (i.e., 00 UTC) between consecutive days, providing two independent positions $\mathbf{r}$ (expressed in meters) at a single epoch for each satellite. The differences between these positions are called orbit midnight discontinuities, as they represent jumps between two smooth orbit arcs. The discontinuity $\delta\mathbf{r}$ in the orbit position between consecutive days has components along $(\delta r_a)$ and cross $(\delta r_c)$ the orbit direction and a radial $(\delta r_r)$ component in the local vertical. Running GROOPS using only *a priori* models (i.e., not estimating any corrections through fitted parameters according to Eq. 4) and analyzing the discontinuities we can compare the goodness of the models – the smaller the discontinuities are, the better is the *a priori* modelling. The associated performance metric $\|\delta\mathbf{r}\|$ is defined as:

$$\|\delta\mathbf{r}\| = \sqrt{(\delta r_a)^2 + (\delta r_c)^2 + (\delta r_r)^2} \tag{7}$$

for each satellite at each day boundary.

Second, as a part of the orbit determination, the *a priori* model for the troposphere can be corrected by fitting parameters in the least squares process (Eq. 4). The fitted residual zenith wet delay $(\Delta D_{zw})$ and gradient delays $(\Delta G_N$ and $\Delta G_E)$ are indicative of the goodness of the *a priori* model used – the smaller the corrections are, the better is the accuracy of the *a priori* model.

## 3 Results

The following results are aimed to demonstrate how the newly assembled experimental configuration (OpenIFS) performs in relation to a well-established default system (VMF3). According to the performance metrics explained in section 2.4, an analysis over the midnight discontinuities has been carried out, along with a study of the fitted parameters for the month of





December 2016. To add statistical confidence to the results, we have also evaluated the month of June 2017 (Appendix A). The

results are hoped to be indicative of the strengths and weaknesses of the experimental configuration, thus pointing to potential

areas of further development.

### 3.1   Orbit midnight discontinuities

The orbit midnight discontinuities are a good metric to analyze how any factor, for instance tropospheric modelling, affects the

quality of GNSS products. Figure 1a shows a color map where each cell represents the difference in the midnight discontinuity

between the *a priori* default and experiment systems for a satellite over two consecutive days, as measured with $\|\delta \mathbf{r}\|$ (see eq. 7

and expressed in units of $\mathrm{cm}$. Here, blue (red) means that discontinuities in the experimental system are smaller (larger) than

in the default system.

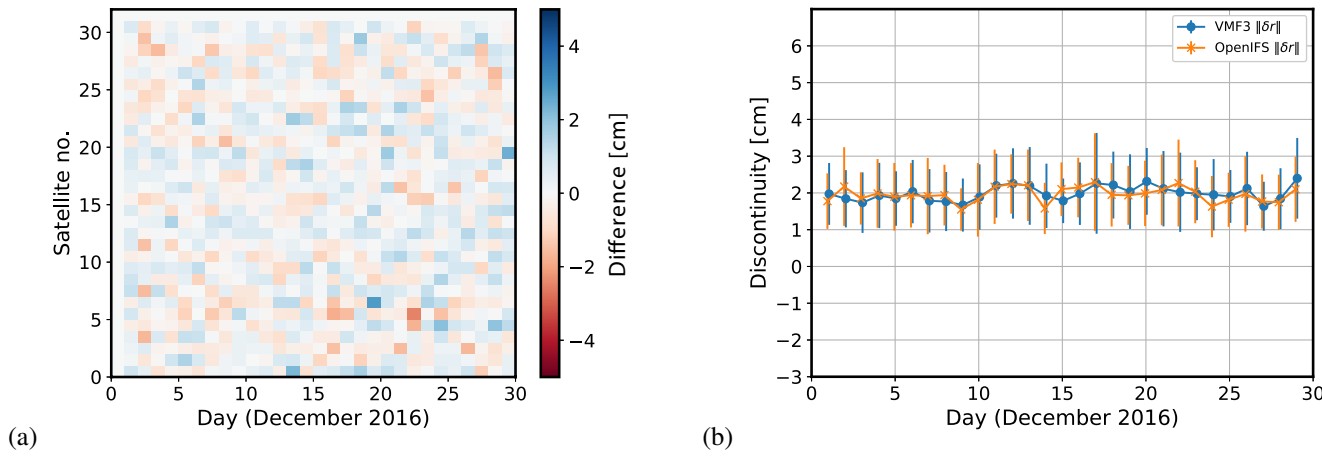

(a)                                                                                                                       (b)

**Figure 1.** Norm of the midnight discontinuity ($\|\delta \mathbf{r}\|$) for December 2016. (a) A color map representing the difference between the default
and experimental system for the satellites used in the experiment; (b) The mean (solid lines) and standard deviation (vertical whiskers) over
all satellites for each day of the experiment with blue (default system) and orange (experimental system).

The differences in Fig. 1a are generally very small, of the order of a few $\mathrm{cm}$. The mean covering all satellites over the whole

period is $0.025\,\mathrm{cm}$, pointing to slightly smaller discontinuities in the experimental system, although statistically negligible.

Next is shown a time-series of the mean values of the norm $\|\delta \mathbf{r}\|$ for all satellites (Fig. 1b), where the default system is in blue

and the experiment in orange. Again, the results indicate that the experimental system has slightly smaller discontinuities. The

mean values for the entire month are $1.961\,\mathrm{cm}$ for the experimental system and $1.986\,\mathrm{cm}$ for the default system. This difference,

albeit small, is a rather systematic feature in Fig. 1b. Standard deviation around the mean value is $0.903\,\mathrm{cm}$ in the experimental

system and $0.907\,\mathrm{cm}$ in the default system, showing a similar behaviour for both systems. The results presented in this section

have been calculated not using Satellite G04 data due to malfunctions in the satellite or satellite-specific processing issues

which are unrelated to tropospheric modelling, that were deteriorating the results for both systems used in the experiment.




## 3.2 Fitted tropospheric parameters

Next, the magnitude of the fitted parameters ($\Delta D_{zw}$, $\Delta G_N$ and $\Delta G_E$) is analyzed. Figure 2a shows the residual zenith wet delay ($\Delta D_{zw}$) for both systems (default in blue, experiment in orange) over the month of December 2016. Each day is represented by the mean and standard deviation of all estimated values for all the stations used.

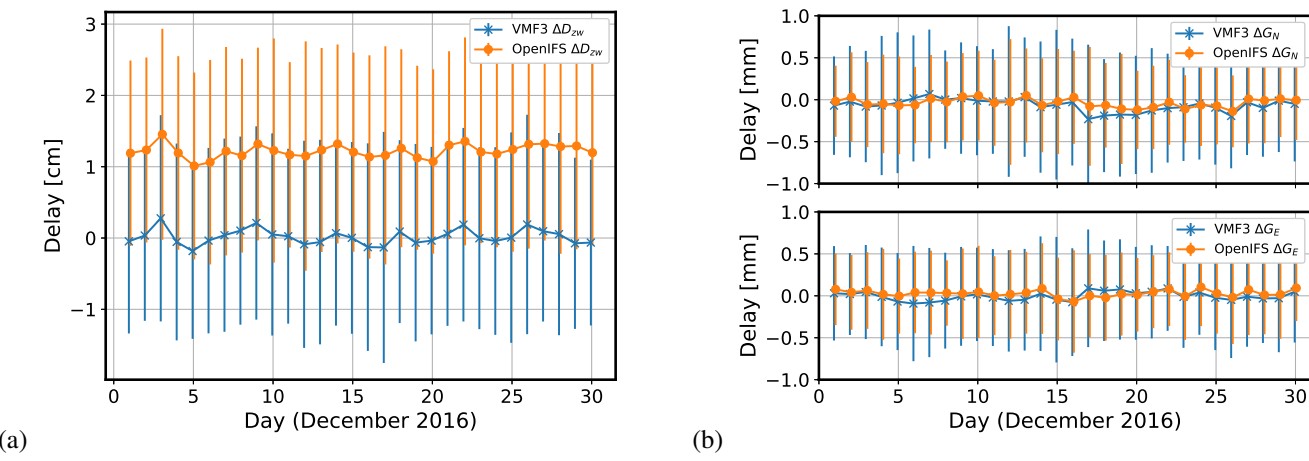

**Figure 2.** The mean and standard deviation of the fitted parameters: (a) zenith wet delay ($\Delta D_{zw}$) and (b) North and East gradient delays ($\Delta G_N$ on top and $\Delta G_E$ at the bottom) for the a priori models for December 2016. Notation is as in Fig. 1

The mean fitted zenith wet delay (Figure 2a) is $1.220$ cm in the experimental system and $0.016$ cm in the default system. This is indicative of a severe and consistent positive bias in the experimental system while it is negligible for the default system. The positive bias can be interpreted such that in the experimental system the total slant delay is too small (i.e., the increased zenith wet delay compensates this deficiency in the least squares process). A test was carried out by segregating the receiver stations to three zones: Northern Hemisphere, Southern Hemisphere and the Tropics. This test does not reveal the sources of the positive bias, and the issue is further discussed in Section 4.

The standard deviation of fitted zenith wet delay values (Figure 2a) is $1.391$ cm in the experimental system and $1.349$ cm in the default system over the whole month. This is indicative of somewhat similar consistency of the default and experimental systems - despite the apparent bias, the variation of the fitted zenith wet delay values is similar to the default system.

The fitted gradient delays in the North and East directions are presented in Fig. 2b. The mean values over the month of December 2016 for $\Delta G_N$ and $\Delta G_E$ are very similar in both systems, $-0.035$ mm and $0.028$ mm for the experiment, and $-0.067$ mm and $-0.003$ mm for the default system. The standard deviations around these mean values are somewhat smaller in the experimental system, $0.527$ mm and $0.489$ mm for the experiment, and $0.706$ mm and $0.602$ mm for the default system, suggesting a marginal advantage over the default system.





### 3.3 Statistical significance of the experimental results

In order to add confidence on the results presented here, the experiment was repeated for the month of June 2017. This period covers a different phase of the Earth on its orbit around the sun, and thus generally different circumstances for the orbit problem. The central results are presented in Appendix A, and can be summarized as being very similar compared to the main study period. The conclusion is thus that the main study period of December 2016 seems to be representative for this type of studies.

## 4 Discussion

The default system of this study applies VMF3 delays and mapping function coefficients, which are computed from the ECMWF weather forecasts. The operational model version in December 2016 was very high-resolution, about 8 km grid spacing at 137 model levels. The disseminated forecast fields for VMF3 computation are however interpolated from the full model resolution to one by one degree horizontal resolution at 25 standard pressure levels, which are available at six-hourly intervals. Lastly, VMF3 coefficients can be interpreted as a low-order representation of the tropospheric delay which does not represent, by design, azimuthal asymmetries in a receiver station sky-view. Despite these processing features, which all imply some loss of atmospheric information, VMF3 has proven to provide a hard-to-beat benchmark also in this study.

The experimental system applies a lower-resolution OpenIFS model version with about 31 km grid spacing at 91 levels, which rather closely corresponds to the system applied in ensemble prediction at ECMWF. In this respect, the default system has a clear advantage in terms of simulation accuracy of atmospheric dynamics and physical processes. On the other hand, the subsequent processing in the experimental system is almost loss-less. We do interpolate the total tropospheric delay from a one by one degree sky-view to azimuth and elevation angles of an individual measurement, and interpolate in time from one-hourly model output fields to the measurement time. Other than that, the native OpenIFS refraction field is represented as such through the ray tracing step to the orbit solver. Technically, the ray tracing could be solved separately and directly corresponding to the azimuth and elevation angles of each measurement – in this case, time interpolation should also be omitted by gathering the measurements into time-slots closest to the model output time. Note that the OpenIFS model time step is 15 minutes in the experimental system and atmospheric fields could be output at the same frequency to reduce time interpolation effects.

Figure 2a is indicative of a sizable bias in the tropospheric *a priori* correction based on OpenIFS, where $\Delta D_{zw}$ is at the level of 0.73 cm throughout the period. This result prompted us to investigate more closely the total delay mean differences between VMF3 and OpenIFS. Indeed, at very low elevation angles the total delay in the experimental system differs systematically from the default system. This is very likely due to an approximation made in the LTT code regarding the geometrical effect on the signal path and the term $\frac{\cos\theta}{nr}(\frac{\partial n}{\partial \psi})_r$ in Eq. 5c which is missing. Therefore, the ray tracer applied in the experimental system does not fully follow Rodgers (2000). Work is ongoing to update the code. Importantly, the experimental system provides us with a solid reference where the weather model-based data is directly interfaced with GNSS measurements and orbit solutions with no intermediate and possibly obscuring processing steps. Thanks to this property, it led us to detect the problem with the implemented ray tracing code.





The following question remains. Figure 2 tends to suggest that the tropospheric *a priori* model in the experimental system is biased. How is it then possible that midnight discontinuities (Fig. 1) are comparable or even slightly smaller in the experimental system compared to the default system? The most likely explanation is that the azimuthal asymmetries of the tropospheric

delay that are present in the experimental system but are missing from the default system do matter and contribute to the orbit solutions. Essentially, the asymmetries are not systematic but they can systematically impact the system performance and the metrics presented here.

Finally, we have to keep in mind that only a small network of 66 stations are included in this study over a limited time period. A more comprehensive experiment will be prepared later including better GNSS station coverage and enhanced tropospheric

modelling. Our hypothesis is that in such a system, more atmospheric information is introduced in near-native format to the orbit problem and it improves the solution accuracy.

## 5   Conclusions

Troposphere and stratosphere delay the propagation of navigation satellite signals and can lead to large errors in GNSS satellite orbit determination if not properly accounted for. In this article, a global numerical weather prediction model – OpenIFS of

260 the European Centre for Medium-Range Weather Forecasts – is applied to generate atmospheric data of pressure, temperature, and humidity. These are converted to 3-dimensional atmospheric refraction fields and used as an input for ray-tracing to solve the least travel time signal paths from satellites to receiver stations and to compute the associated delays. These delays are then directly used as *a priori* corrections of the atmospheric effects to solve GNSS satellite orbits with the GROOPS software toolkit of Graz University of Technology, Austria.

This new configuration to solve the GNSS orbit problem contains two novel aspects. First, the direct use of tropospheric delays on slanted signal paths allows to fully account for the azimuthal asymmetries in the atmospheric refraction field, in contrast to traditional mapping functions which regularize the refraction field. Second, the intimate coupling of the numerical weather prediction model with the orbit solver allows to control the information flow between the two modules, including output frequency of the weather model, for instance.

The main finding here is that the new configuration provides good consistency of GNSS satellite orbits as measured with the so-called orbit midnight discontinuities, i.e., how much satellite orbit initial positions need to be corrected between subsequent 24-hour orbit solutions. Another important finding is that the new configuration provides a solid reference where weather model-based data is directly interfaced with GNSS measurements and orbit solutions. Since the new configuration has fewer intermediate processing steps than the state-of-the-art methods, more direct diagnosis of the system performance is possible. In

particular, this led us to detect a bias in our ray tracing computation that was undetected until now. Finally, the results indicate that azimuthal asymmetries in the tropospheric delays, which are well-preserved in the new configuration, contribute to the accuracy of orbit solutions. These asymmetries are not systematic but they have a systematic impact on the orbit solutions.

At a more general level, the new configuration provides an enhanced control of geodetic and meteorological aspects of the orbit problem. This will allow us to widely benefit the apparent synergies in space geodesy and meteorology.





## Appendix A: The experiment using June 2017 data

The experimental results for the month of June 2017 are presented here. The experimental setup is identical to the experiment covering the month of December 2016. The central results, following the presentation of Section 2.4, are shown below.

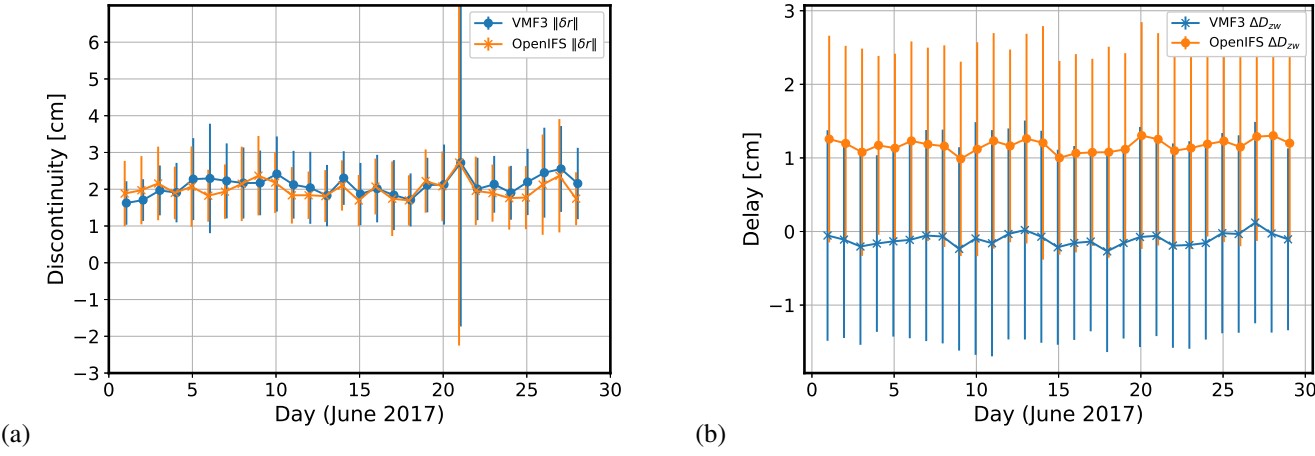

**Figure A1.** Daily means of (a) the midnight discontinuities and (b) the fitted zenith wet delays ($\Delta D_{zw}$) of the default model (blue) and the experiment model (orange) for the month of June 2017.

Figure A1a can be compared to Fig. 1b, showing the daily mean of the midnight discontinuities of the *a priori* systems for the month. The overall means and standard deviations for the months of December 2016 and June 2017 are: $(1.961 \pm 0.903)$ cm and $(1.996 \pm 1.028)$ cm for the experimental system and $(1.986 \pm 0.907)$ cm and $(2.102 \pm 1.034)$ cm for the default system.

Figure A1b can be compared to Fig. 2a, showing the fitted parameters of the zenith wet delay ($\Delta D_{zw}$). The bias in the experimental system for December 2016 (Fig. 2a) appears to be at a similar level also in June 2017. The overall mean and standard deviation in the experimental system is $(1.220 \pm 1.391)$ cm for December 2016 and $(1.168 \pm 1.371)$ cm for June 2017. On the other hand, the negligible bias ($10^{-2}$ cm) in the default system in December 2016 seems to be slightly moved to negative for June 2017, although the overall standard deviation is very similar: $(0.016 \pm 1.349)$ cm for December 2016 and $(-0.108 \pm 1.370)$ cm for June 2017.

*Code availability.* The source code of GROOPS is openly available on GitHub (https://github.com/groops-devs/groops). Licence for using OpenIFS NWP model can be requested from ECMWF user support (openifs-support@ecmwf.int), and the model can be downloaded from the ECMWF FTP site (ftp.ecmwf.int). The LTT-operator used in this study is available on Zenodo under Creative Commons Attribution 4.0 International licence (https://doi.org/10.5281/zenodo.4834411; Eresmaa et al. 2021).



*Data availability.* Availability of the OpenIFS initial states is described in Ollinaho et al. (2021). GNSS observation data are kindly provided by the IGS via its data centers (https://igs.org/data-products-overview).

*Author contributions.* ANT performed experimentation and most of the data analysis, SS prepared and supervised the experimentation, LT ran the OpenIFS model and post-processed slant delay data, MV made the slant delay sensitivity analyses, maintained the LTT operator,
and participated in production of the slant delay data, MP supported development of the concept, TMG provided in-depth expertise in the experimental design and supervised in all aspects of the project, and HJ supervised in all aspects of the project as its originator. All authors contributed to writing with coordination of HJ.

*Competing interests.* There are no competing interests.

*Acknowledgements.* We gratefully acknowledge support of the Doctoral Programme in Atmospheric Sciences in the University of Helsinki,
and funding received from the Academy of Finland (grant number 1333034), the Vilho, Yrjö and Kalle Väisälä Foundation, and the Austrian Research Promotion Agency (FFG) under the Austrian Space Applications Programme (ASAP) Phase 16 (project number 878886). We would like to thank CSC - IT Center for Science in Finland - for their support in high-performance computing. Juha Lento from CSC is warmly acknowledged for his support and advice in implementing GROOPS in a massively parallel computing environment.





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
