# Peer review of "Coupling a weather model directly to GNSS orbit determination - case studies with OpenIFS"

_Geoscientific Model Development, 2021_

## Author Comment (AC2)

**Major recommendation 1**

>>>>*Line 205ff: "A test was carried out by segregating the receiver stations to three zones: Northern Hemisphere, Southern Hemisphere and the Tropics. This test does not reveal the sources of the positive bias ...". As reader of the manuscript, I am not very happy that the analysis stops at this point. I would have expected that the developed ray-tracer is compared to other existing ray-tracers, e.g. as provided by TU Vienna: https://vmf.geo.tuwien.ac.at/. On line 240 it is concluded that the positive bias "is very likely due to an approximation made in the LTT code". However, no justification for this statement is provided in the manuscript yet. In addition or as alternative, the forecasted refraction field from the OpenIFS system could have been compared to operational forecast data to make clear whether the positive bias is caused by ray-tracing or the input field.*

We want to thank the referee for this recommendation specifically, as although the recommendation is centainly valid, this paper was meant to introduce the possibility of directly couple both the Precise Orbit Determination Solver (GROOPS) and the Numerical Weather Prediction Model (OpenIFS), in this case using the ray-tracer of Least Travel Time (LTT). Due to this, we agree on the lack of necessity to add the test with segregated stations, since this is part of a different study and only adds confusion to the reader, thus this sentence has been removed. Once the couple of OpenIFS and GROOPS had been proven, a second study comparing the suggested ray-tracer and the LTT started developing, leading to the improvement of the LTT ray-tracer and to the prove of the validity of such ray-tracer.

Respect to the second part of the recommendation, we would like to appreciate this comment from the referee as we did not notice that the explanation was not given in the introduction, although it was given in Section 2.3 (Slant Delays), regarding our implementation. We have added a brief explanation in the introduction and extended the explanation in Section 2.3, so that now the wording on line 240 has a reference to these explanations.

**Major recommedations 2**

>>>>*Line 250f: The statement that the "azimuthal asymmetries of the tropospheric delay that are present in the experimental system but are missing from the default system do matter and contribute to the orbit solutions" is not supported by the data. The same is valid for line 277f. Please provide the corresponding justification, i.e. by repeating the analysis without azimuthal asymmetries (remove them from the ray-traced delays priori to GNSS data processing).*

The main difference between the experimental system and the default system relies in the production of the skyviews. The default system uses a map function that depends on the elevation angle and three constants that change only every certain time, thus it has no variation over the azimuth angle, losing valuable information. In the experimental system the skyview is produced including the azimuth component, so the delays vary with respect the azimuth angle. This effect is expect to be occuring in the skies as the troposphere varies a lot. Including these 'asymmetries' leads to an improvement of the tropospheric model and removing them would mean the elimination of the main difference between both systems. Even though we are trying to show the possibility of coupling a Weather Model to a Precise Orbit Determination Solver,  we also would like to show the importance on these asymmetries in the calculation of the slant delays and their posterior effect in orbit determination.

**Minor recommedation 1**

>>>>*Lines 1-6: This part is a bit confusing. I suggest: "Neutral gas atmosphere bends and delays propagation of microwave signals in satellite-based navigation. Weather prediction models can be*

*used to estimate these effects by providing 3-dimensional refraction fields to ray-trace the signal delays. In this study, a global numerical weather prediction model (Open Integrated Forecasting System (OpenIFS) licensed for Academic use by the European Centre for Medium-RangeWeather Forecast) is used to generate the refraction fields. The ray-traced slant delays are supplied as such for an orbit solver (GROOPS (Gravity Recovery Object Oriented Programming System) software toolkit of the Technical University of Graz) which applies the raw observation method.*

We agree with the referee suggestion, as the modifications improve the text and makes it easier for the reader.

**Minor recommendation 2**

*>>>>Lines 10-11: Remove "as measured with the midnight discontinuity of Global Navigation Satellite System (GNSS) satellite orbits" since it is not relevant at this point.*

We agree with the referee, and that line will be removed from the text.

**Minor recommendation 3**

*>>>>Line 14: What is meant by "precision-reducing mapping"? Please rephrase.*

This part has not been correctly explained, and some text needs to be added. "Moreover, the direct coupling helps in identifying deficiencies in the slant delay computation because the modelling errors are not convoluted in the mapping functions, which can reduce the precision due to the loss of information."

**Minor recommendation 4**

*>>>>Line 155: "Here the LTT solver is applied such that instead of computing ray paths exactly in direction of the GNSS satellites in view, so-called sky-views are generated."*

We agree with the suggestion made by the referee, and the modifications will be applied to the text.

---

## Author Comment (AC3)

**Abstract recommendations**

>>>>>*On first reading my impression was that the use of openIFS is the novel thing, but on a second reading, remembering what I have read before, my opinion changed. Now I believe that it is the use of the ray-tracing instead of the mapping functions that is the new aspect here. I suggest that this should be made clearer so that the reader gets it on the first reading.*

We would like to thank the referee for pointing this out. After some debate we consider that the abstract focuses on the use of OpenIFS data as the novel thing in the coupling of weather models and orbit determination solvers, as it was intended. The use of the ray tracer allows us to bypass the creation of the mapping function, that will be the continuation of this work, but we could use RADIATE (ray tracer used by Vienna to create their mapping functions) to get to the same results, although losing some information due to the mapping functions.

>>>>>*I am also a bit surprised that the whole paper deals with a-priori estimates only. Perhaps it is my ignorance of this field, but why do you compare only a-priori estimates instead of final ones?*

Thanks to the referee for this question, as it is important to understand the reason to using 'a priori' estimates and it has not been clearly explained. From the perspective of GNSS orbit determination, the troposphere is a major source of perturbations of the GNSS signals. Hence, we try to remedy this by modeling the signal delay caused by the troposphere so we can subtract/remove it from the observations. This is what we call the "a priori" troposphere model. Unfortunately, these models are so far not able to represent the wet component of the troposphere sufficiently well. Therefore, we have to estimate certain tropospheric parameters during the orbit determination process that account for the remaining (unmodeled) wet tropospheric signal delay. These are referred to as a posteriori. In short, comparing a-priori estimates is the most straight-forward method to assess the performance of the two ways to process the input data for precise orbit determination.

>>>>>*A minor issue was the notion "midnight discontinuity" which was puzzling on first reading. Later it became clear. Please try to find a short but simple circumscription of what is meant so that the reader does not get lost already in the abstract.*

The referee is right about this issue and we appreciate the notification as it is important to state what we meant with 'midnight discontinuity'. A comment from another referee suggested to remove the sentece and, after some consideration, we decided to remove it, so this 'minor recommendation' needs no further revision.

**Section 2.1 recommendations**

>>>>>*It is astonishing to me that 30 sec of data per day are sufficient to compute the orbit for a complete 24 hours period. Is this, because it is essentially celestial mechanics (i.e. almost negligible perturbations) or is there a misunderstanding?*

We think that there has been a misunderstanding here. The observation samplig is done in epochs of 30 seconds, so over one day we have 2880 observation epochs with which we compute the orbits. The perturbations are in fact important. We assume the responsibility of the misunderstanding and we would like to thank the referee for noticing this error.

>>>>>*What do you mean with "pseudostochastic pulses are estimated"? Don't repeat the quoted paper here, but a few simple words for explantion would be fine.*

As the perturbations of the satellites are important and the modelling is not perfect, an additional 'force' needs to be added as a correction for these deficiencies in the orbit. This force is included as an increment of the velocity of the satellite in the center of the 24-h period. As the force is not a real factor and has just been introduced to make a correction, we call it pseudostochastic.

>>>>>*I understand that you are computing a-priori delays, but in lines 99 ff you compare them with actual delays and the difference or discrepancy is determined (or estimated). To me it is unclear from where you have the actual delays and whether you need an a-priori estimate at all if you have the actual delay. Or is the a priori for orbit prediction and the actual positions can be measured after the event?*

This question is directly related to 'Abstract recommendations, issue number 2'. In this paragraph of the manuscript (lines 99 ff) the explanation of the impact of the troposphere is explained. As all the other factors in the orbit determination process are well modelled, the discrepancies over the slant delays are assumed to come from the troposphere measurement and thus we can estimate how large the discrepacies are. The tropospheric delay is divided in two parts, dry and wet. Whereas the dry part is well modelled, the wet part is not so easy to model as it changes rapidly, so the discrepancies of the slant delays can be attached to this second part, the wet delay. Even though the objective is to get rid of the a-posteriori calculations, we can actually use them now to see deficiencies in our model (OpenIFS) and our ray-tracer (LTT).

The actual tropospheric signal delay affecting a GNSS observation is not known during orbit determination. GNSS signals are affected by several perturbations (ionosphere, tropsphere, signal biases etc.). Each perturbation is modeled as well as possible. The remaining discrepancy between the measured total signal delay and the modeled signal delay can then be analyzed with respect to unmodeled effects. For example, it can tell us about un-/mismodeled tropospheric effects.

**Section 3.2 recommendation**

>>>>>*I wonder why the Deltas (to my understanding, the difference between the a-priori estimate and the actual delays) are compared for the two methods instead of the a-priori parameters directly. It should make no difference, of course, since the actual delay must be the same for both methods and must thus cancel out. But it is puzzling. I would expect instead two tests here: 1) the comparison of both methods using the a-priori parameters only and 2) the distribution of the deltas in the new version, to see whether the difference of the two methods is significant in comparison to the difference between model and measurement. While the first of these steps are given, the second is missing and should be given.*

We appreciate this question from the referee as the procedure is tricky to understand and we may have not explained it correctly. In this section the fitted parameters, that are the differences obtained from the measurements and the total delay, are compared for both systems, experimental and default (explained in previous replies). The discrepancies between measurements and real delays come from mathematical methods, and are assummed to come solely from the troposphere. Knowing these discrepancies and extending them to the fitted equation posted in Section 2.1, we can get an idea of where these discrepancies are being produced (the wet component or the gradients) and spot deficiencies in the model and or data.

**Section 4 recommendation**

>>>>>*For the VMF3 method you use the operational weather forecast, and for the new method the openIFS. Your interest is not to demonstrate the differences arising from using one set of weather data that has millions of observations assimilated with another data set without data assimilation. Instead your interest is to compare the use of the ray-tracing method with the mapping method, if I undestand it correctly. But of course, the atmospheric states should be different in the two weather data sets, and the one using the actual forecast should be more realistic. So it is surprising that this does not lead to a significantly better performance of the mapping method. I would like to see your comments on this issue.*

Although OpenIFS itself does not have the data assimilation capability, the observation data has already been assimilated beforehand during the creation of the initial condition files for OpenIFS. Similarly, in the creation of the Vienna mapping function coefficients, the observation data assimilation is done beforehand during the operational forecast production. VMF3 then uses the forecasts and the RADIATE ray tracer to produce their mapping function coefficients. However, we are trying to avoid some steps by directly using the ray tracer over OpenIFS data. The atmospheric states are somehow different in each forecast system, but in this manuscript the idea was coupling the weather model to the orbit determination solver as we expect to improve our results. In fact a new study is being carried out to analyze the improvements in both, the ray tracer and the orbit determination solver, with outstanding results that are leading to a new paper.

---

## Author Response (AR2)

**REFEREE 1 RESPONSE**

**Major recommendation 1**

>>>>*Line 205ff: "A test was carried out by segregating the receiver stations to three zones: Northern Hemisphere, Southern Hemisphere and the Tropics. This test does not reveal the sources of the positive bias ...". As reader of the manuscript, I am not very happy that the analysis stops at this point. I would have expected that the developed ray-tracer is compared to other existing ray-tracers, e.g. as provided by TU Vienna: https://vmf.geo.tuwien.ac.at/. On line 240 it is concluded that the positive bias "is very likely due to an approximation made in the LTT code". However, no justification for this statement is provided in the manuscript yet. In addition or as alternative, the forecasted refraction field from the OpenIFS system could have been compared to operational forecast data to make clear whether the positive bias is caused by ray-tracing or the input field.*

We want to thank the referee for this recommendation specifically, as although the recommendation is centainly valid, this paper was meant to introduce the possibility of directly couple both the Precise Orbit Determination Solver (GROOPS) and the Numerical Weather Prediction Model (OpenIFS), in this case using the ray-tracer of Least Travel Time (LTT). Due to this, we agree on the lack of necessity to add the test with segregated stations, since this is part of a different study and only adds confusion to the reader, thus this sentence has been removed. Once the couple of OpenIFS and GROOPS had been proven, a second study comparing the suggested ray-tracer and the LTT started developing, leading to the improvement of the LTT ray-tracer and to the prove of the validity of such ray-tracer.

Respect to the second part of the recommendation, we would like to appreciate this comment from the referee as we did not notice that the explanation was not given in the introduction, although it was given in Section 2.3 (Slant Delays), regarding our implementation. We have added a brief explanation in the introduction and extended the explanation in Section 2.3, so that now the wording on line 240 has a reference to these explanations.

A second phrase has been added to the end of the *Introduction* section to state that the study of the ray-tracer is out of the scope of this paper, and also to line 158.

**Major recommedations 2**

>>>>*Line 250f: The statement that the "azimuthal asymmetries of the tropospheric delay that are present in the experimental system but are missing from the default system do matter and contribute to the orbit solutions" is not supported by the data. The same is valid for line 277f. Please provide the corresponding justification, i.e. by repeating the analysis without azimuthal asymmetries (remove them from the ray-traced delays priori to GNSS data processing).*

We appreciate the recommendation of the referee, as we can not yet claim that the azimuthal asymmetries do improve the results. Thus, we would like to withdrawn the claim. There have been three modifications to the manuscript in order to correct this issue: line 10, line 30 and line 249.

We would like to explain the reasoning to our suspicion.

The main difference between the experimental system and the default system relies in the production of the skyviews. The default system uses a map function that depends on the elevation angle and three constants that change only every certain time, thus it has no variation over the azimuth angle, losing valuable information. In the experimental system the skyview is produced including the azimuth component, so the delays vary with respect the azimuth angle. This effect is

expect to be occuring in the skies as the troposphere varies a lot. Including these 'asymmetries' leads to an improvement of the tropospheric model and removing them would mean the elimination of the main difference between both systems. Even though we are trying to show the possibility of coupling a Weather Model to a Precise Orbit Determination Solver, we also would like to show the importance on these asymmetries in the calculation of the slant delays and their posterior effect in orbit determination.

**Minor recommedation 1**

>>>>*Lines 1-6: This part is a bit confusing. I suggest: "Neutral gas atmosphere bends and delays propagation of microwave signals in satellite-based navigation. Weather prediction models can be used to estimate these effects by providing 3-dimensional refraction fields to ray-trace the signal delays. In this study, a global numerical weather prediction model (Open Integrated Forecasting System (OpenIFS) licensed for Academic use by the European Centre for Medium-RangeWeather Forecast) is used to generate the refraction fields. The ray-traced slant delays are supplied as such for an orbit solver (GROOPS (Gravity Recovery Object Oriented Programming System) software toolkit of the Technical University of Graz) which applies the raw observation method.*

We agree with the referee suggestion, as the modifications improve the text and makes it easier for the reader.

**Minor recommendation 2**

>>>>*Lines 10-11: Remove "as measured with the midnight discontinuity of Global Navigation Satellite System (GNSS) satellite orbits" since it is not relevant at this point.*

We agree with the referee, and that line will be removed from the text.

**Minor recommendation 3**

>>>>*Line 14: What is meant by "precision-reducing mapping"? Please rephrase.*

This part has not been correctly explained, and some text needs to be added. "Moreover, the direct coupling helps in identifying deficiencies in the slant delay computation because the modelling errors are not convoluted in the mapping procedures."

**Minor recommendation 4**

>>>>*Line 155: "Here the LTT solver is applied such that instead of computing ray paths exactly in direction of the GNSS satellites in view, so-called sky-views are generated."*

We agree with the suggestion made by the referee, and the modifications will be applied to the text.

**REFEREE 2 RESPONSE**

**Abstract recommendations**

>>>>>*On first reading my impression was that the use of openIFS is the novel thing, but on a second reading, remembering what I have read before, my opinion changed. Now I believe that it is*

*the use of the ray-tracing instead of the mapping functions that is the new aspect here. I suggest that this should be made clearer so that the reader gets it on the first reading.*

We would like to thank the referee for pointing this out. After some debate we consider that the abstract focuses on the use of OpenIFS data as the novel thing in the coupling of weather models and orbit determination solvers, as it was intended. The use of the ray tracer allows us to bypass the creation of the mapping function, that will be the continuation of this work, but we could use RADIATE (ray tracer used by Vienna to create their mapping functions) to get to the same results, although losing some information due to the mapping functions.

>>>>>*I am also a bit surprised that the whole paper deals with a-priori estimates only. Perhaps it is my ignorance of this field, but why do you compare only a-priori estimates instead of final ones?*

Thanks to the referee for this question, as it is important to understand the reason to use 'a priori' estimates and it has not been clearly explained. From the perspective of GNSS orbit determination, the troposphere is a major source of perturbations of the GNSS signals. Hence, we try to remedy this by modeling the signal delay caused by the troposphere so we can subtract/remove it from the observations. This is what we call the "a priori" troposphere model. We do work with these "a priori" models in the first performance metric (figure 1). Unfortunately, these models are so far not able to represent the wet component of the troposphere sufficiently well. Therefore, we have to estimate certain tropospheric parameters during the orbit determination process that account for the remaining (unmodeled) wet tropospheric signal delay. These are referred to as a posteriori. This is shown in figure 2 of the manuscript, where we use the 'a posteri' models. In short, comparing a-priori estimates is the most straight-forward method to assess the performance of the two ways to process the input data for precise orbit determination. We then have used both models, a-priori and 'final' ones in the manuscript.

>>>>>*A minor issue was the notion "midnight discontinuity" which was puzzling on first reading. Later it became clear. Please try to find a short but simple circumscription of what is meant so that the reader does not get lost already in the abstract.*

The referee is right about this issue and we appreciate the notification as it is important to state what we meant with 'midnight discontinuity'. A comment from another referee suggested to remove the sentece and, after some consideration, we decided to remove it, so this 'minor recommendation' needs no further revision.

**Section 2.1 recommendations**

>>>>>*It is astonishing to me that 30 sec of data per day are sufficient to compute the orbit for a complete 24 hours period. Is this, because it is essentially celestial mechanics (i.e. almost negligible perturbations) or is there a misunderstanding?*

We think that there has been a misunderstanding here. The observation samplig is done in epochs of 30 seconds, so over one day we have 2880 observation epochs with which we compute the orbits. The perturbations are in fact important. We assume the responsibility of the misunderstanding and we would like to thank the referee for noticing this error. We have added the number of observations in the text, line 66.

>>>>>*What do you mean with "pseudostochastic pulses are estimated"? Don't repeat the quoted paper here, but a few simple words for explantion would be fine.*

As the perturbations of the satellites are important and the modelling is not perfect, an additional 'force' needs to be added as a correction for these deficiencies in the orbit. This force is included as an increment of the velocity of the satellite in the center of the 24-h period. As the force is not a real factor and has just been introduced to make a correction, we call it pseudostochastic.

The explanation has been improved in the text (line 72) to help the reader.

>>>>>*I understand that you are computing a-priori delays, but in lines 99 ff you compare them with actual delays and the difference or discrepancy is determined (or estimated). To me it is unclear from where you have the actual delays and whether you need an a-priori estimate at all if you have the actual delay. Or is the a priori for orbit prediction and the actual positions can be measured after the event?*

This question is directly related to 'Abstract recommendations, issue number 2'. In this paragraph of the manuscript (lines 99 ff) the explanation of the impact of the troposphere is explained. As all the other factors in the orbit determination process are well modelled, the discrepancies over the slant delays are assumed to come from the troposphere measurement and thus we can estimate how large the discrepacies are. The tropospheric delay is divided in two parts, dry and wet. Whereas the dry part is well modelled, the wet part is not so easy to model as it changes rapidly, so the discrepancies of the slant delays can be attached to this second part, the wet delay. Even though the objective is to get rid of the a-posteriori calculations, we can actually use them now to see deficiencies in our model (OpenIFS) and our ray-tracer (LTT).

The actual tropospheric signal delay affecting a GNSS observation is not known during orbit determination. GNSS signals are affected by several perturbations (ionosphere, tropsphere, signal biases etc.). Each perturbation is modeled as well as possible. The remaining discrepancy between the measured total signal delay and the modeled signal delay can then be analyzed with respect to unmodeled effects. For example, it can tell us about un-/mismodeled tropospheric effects.

**Section 3.2 recommendation**

>>>>>*I wonder why the Deltas (to my understanding, the difference between the a-priori estimate and the actual delays) are compared for the two methods instead of the a-priori parameters directly. It should make no difference, of course, since the actual delay must be the same for both methods and must thus cancel out. But it is puzzling. I would expect instead two tests here: 1) the comparison of both methods using the a-priori parameters only and 2) the distribution of the deltas in the new version, to see whether the difference of the two methods is significant in comparison to the difference between model and measurement. While the first of these steps are given, the second is missing and should be given.*

We appreciate this question from the referee as the procedure is tricky to understand and we may have not explained it correctly. In this section the fitted parameters, that are the differences obtained from the measurements and the total delay, are compared for both systems, experimental and default (explained in previous replies). The discrepancies between measurements and real delays come from mathematical methods, and are assumed to come solely from the troposphere. Knowing these discrepancies and extending them to the fitted equation posted in Section 2.1, we can get an idea of where these discrepancies are being produced (the wet component or the

gradients) and spot deficiencies in the model and or data. Thus, the second step proposed is indeed the second metric used in the manuscript to explain the differences between the experimental and default models (figure 2).

**Section 4 recommendation**

>>>>>*For the VMF3 method you use the operational weather forecast, and for the new method the openIFS. Your interest is not to demonstrate the differences arising from using one set of weather data that has millions of observations assimilated with another data set without data assimilation. Instead your interest is to compare the use of the ray-tracing method with the mapping method, if I undestand it correctly. But of course, the atmospheric states should be different in the two weather data sets, and the one using the actual forecast should be more realistic. So it is surprising that this does not lead to a significantly better performance of the mapping method. I would like to see your comments on this issue.*

Although OpenIFS itself does not have the data assimilation capability, the observation data has already been assimilated beforehand during the creation of the initial condition files for OpenIFS. Similarly, in the creation of the Vienna mapping function coefficients, the observation data assimilation is done beforehand during the operational forecast production. VMF3 then uses the forecasts and the RADIATE ray tracer to produce their mapping function coefficients. However, we are trying to avoid some steps by directly using the ray tracer over OpenIFS data. The atmospheric states are somehow different in each forecast system, but in this manuscript the idea was coupling the weather model to the orbit determination solver as we expect to improve our results. In fact a new study is being carried out to analyze the improvements in both, the ray tracer and the orbit determination solver, with outstanding results that are leading to a new paper.

Some text has been modified in the first paragraph of section 2.2 Weather Model to address the data assimilation.

**LIST OF CHANGES**

**Line 3:**
- Removed 'estimate signal delay in the zenith direction and determine a low-dimensional mapping of this delay to desired azimuth and elevation angles' .
- Added 'ray-trace the signal delays'.

**Line 11:**
- Removed ', as measured with the midnight discontinuity of Global Navigation Satellite System (GNSS) satellite orbits'
- Removed 'Our interpretation is'
- Added 'We suspect'

**Line 14:**
- Removed 'precision reducing mapping'
- Added 'mapping procedures'

**Line 39:**
- Added 'To this end, we have first analyzed the performance of tropospheric delays produced by both systems (VMF3 and OpenIFS), providing the so called 'a priori' models. Then we have analyzed the differences of these two models in a real case, by correcting the \emph{a priori} model by fitting parameters in a least squares process'.

**Line 29:**
-Removed 'parameters and the azimuthally asymmetric part with two additional parameters'
-Added 'mapping coefficients, and has a dependency to the elevation angle but not to the azimuthal angle'.

**Line 49:**
- Added 'In this study, we have used OpenIFS at an affordable resolution (T$_\mathrm{L}$639L91, see section 2.2) and a version of the least traveltime (LTT) ray-tracer used in~\citet{Eresmaa_2008}. We acknowledge that this setup does not utilize the full potential of production of slant-path corrections yet. Firstly, the LTT ray-tracer does not fully follow~\citet{Rodgers_2000} (see section 2.3), and secondly, the resolution of OpenIFS can be increased further. However, even though a study and improvement of the ray-tracer could be done, it is out of the scope of this paper and will be addressed in an upcoming publication, as this setup allows us to experiment with the new configuration'.

**Line 68:**
- Added '(a total of 2880 observation epochs per day)'

**Line 78:**
- Added 'called pseudo-stochastic pulses as they are not physical forces but corrections to deficiencies in the calculations'

**Line 119:**
- Removed 'and thus' , 'required'.
- Added 'explicitely' , 'but it is included implicitly because' , 'applied'.

**Line 158:**
- Added ' This implementation does not completely follow~\citet{Rodgers_2000} and therefore the results are sub-optimal. Efforts are ongoing to improve the LTT implementation, but this is out of the scope of this paper and it will be addressed in a future publication.'

**Line 165:**
- Removed 'individual GNSS measurement geometries'
- Added 'in direction of the GNSS satellites in view'

**Line 182:**
- Added ', producing the so-called \textit{a posteriori} model.

**Line 183:**
- Added 'The reason to deal with \emph{a prior} models is that, if this model would be perfect, there would be no need to add more unknowns to the least squares, reducing the amount of noise and speeding the calculations. When adding more variables to the least squares, we may dilute errors from different sources into the variables, hiding the responsible of such errors. The ideal case would be to not have the need to use these \emph{a posteriori} models.'

**Line 215:**
- Removed 'A test was carried out by segregating the receiver stations to three zones: Northern Hemisphere, Southern Hemisphere and the Tropics. This test does not reveal the sources of the positive bias, and the issue is further discussed in Section 4'
- Added 'The possible reasons for the bias arefurther discussed in Section 4'.

**Line 252-253:**
- Removed 'and', 'which'
- Added 'where', 'LTT' and 'and is sub-optimal'

**EDITOR MINOR RECOMMENDATIONS**

We would like to thank the editor for taking the time to review everything and for pointing these minor issues. We have solved the first two. As for the last one 'Address the responses to referee 2 in the text' could not be fully completed. We have addressed all the responses that were possible, but some of them were just responses to the referee and not specifically to the suggestions, so they could not be included in the manuscript.

The changes in the responses with respect to the previous version have been colored in blue, and in the manuscript in red (as blue was the color from the previous version).